# Bayesian Machine Scientist for Model Discovery in Psychology

**Joshua TS Hewson**
Brown University
joshua_hewson@brown.edu

**Younes Strittmatter**
Brown University
younes_strittmatter@brown.edu

**Ioana Marinescu**
Princeton University
ioanam@princeton.edu

**Chad C Williams***
Brown University
chad_williams@brown.edu

**Sebastian Musslick***
Osnabrück University, Brown University
sebastian.musslick@uos.de

***Co-senior author**

## Abstract

The rapid growth in complex datasets within the field of psychology poses challenges for integrating observations into quantitative models of human information processing. Other fields of research, such as physics, proposed equation discovery techniques as a way of automating data-driven discovery of interpretable models. One such approach is the Bayesian Machine Scientist (BMS), which employs Bayesian inference to derive mathematical equations linking input variables to an output variable. While BMS has shown promise, its application has been limited to a small subset of scientific domains. This study examines the utility of BMS for model discovery in psychology. In Experiment 1, we compare BMS in recovering four models of human information processing against two common psychological benchmark models—linear/logit regression and a black-box neural network—across a spectrum of noise levels. BMS outperformed the benchmark models on the majority of noise levels and demonstrated at least equivalent performance when considering higher levels of noise. These findings demonstrate BMS's potential for discovering psychological models of human information processing. In Experiment 2, we investigated the impact of informed priors on BMS recovery, comparing domain-specific function priors against a benchmark uniform prior. Specifically, we investigated four priors across research domains spanning their specificity to psychology. We observe that informed priors robustly enhanced BMS performance for only one of the four models of human information processing. In summary, our findings demonstrate the effectiveness of BMS in recovering computational models of human information processing across a range of noise levels; however, whether integrating expert knowledge into the BMS framework improves performance remains a subject of further inquiry.

## 1 Introduction

Incorporating behavioral phenomena into models of human information processing is a cornerstone of psychology. However, the task of integrating empirical data into these quantitative models has become increasingly complex due to the rapid growth in both the amount and complexity of available data. This challenge is not unique to psychology; fields like physics, chemistry, and materials science have also grappled with it, leading them to explore automated methods for model discovery [1–4]. A predominant focus in this area has been on equation discovery, also known as symbolic regression [5].

NeurIPS 2023 AI for Science Workshop.

This approach aims to identify mathematical expressions that can accurately relate the input variables of an experiment to the observed outcomes. In this article, we assess the efficacy of using equation discovery to derive psychological models of information processing.

There have been different algorithmic approaches to equation discovery, all of which seek to identify computation graphs that relate input variables to an output variable. Recent approaches to equation discovery include brute-force graph search [6], genetic algorithms [7–9], differentiable architecture search [10], Bayesian inference [11, 12], reinforcement learning [13, 14], and sparse regression [15–18]. Among them is the Bayesian Machine Scientist (BMS) [11], which employs Bayesian inference to update beliefs about model parameters and structures based on observed data. The algorithm systematically evaluates, refines, and compares mathematical expressions, focusing on simplicity and interpretability. BMS has three key advantages: (1) its capability for a prior-informed search that allows for the integration of existing domain expertise, (2) its Occam's razor-inspired approach that favors simplicity, promoting interpretable mathematical models, and (3) its successful application to real-world scientific problems [19–27].

Although BMS has proven effective in deriving quantitative models from data, its application has been confined to a small subset of scientific domains, such as chemical engineering [20, 21, 25–27], systems science [19, 24], and physics [22]. Using BMS, we extend the application of equation discovery to the realm of psychology[1], specifically for recovering mathematical models of human information processing. Our work makes three key contributions:

1. We assess the capability of BMS to accurately reconstruct well-established models of human information processing using synthetic data. This involves a comparative analysis with statistical models commonly employed in psychology to describe the relationships between experimental variables and observed outcomes (Experiment 1).

2. We investigate the impact of noise on the model recovery process, quantifying the results in terms of both mean squared error and the percentage of perfect recoveries (Experiment 1).

3. We evaluate the benefits of domain-specific function priors, sourced from Wikipedia, to enhance the efficacy of BMS in recovering psychological models (Experiment 2).

The code for BMS described in this paper is available as an open-source Python package named *AutoRA-Theorist-BMS*, which is built to be compatible with a larger package named *AutoRA*. Documentation of this package and all code for the simulations can be found *here*. Furthermore, the code for webscraping the priors used in Experiment 2 is available as an open-source Python package named *Equation-Scraper*, and documentation can be found *here*.

## 2  Equation Discovery with the Bayesian Machine Scientist

The experiments described below leverage the Bayesian Machine Scientist as described in Guimerà et al., 2020 [11]. Here, we provide a brief overview of the method.

### 2.1  Expression Trees

BMS represents mathematical expressions as trees (Figure 1A). *Leaf nodes* indicate input variables (e.g., experimental factors) or constants. *Intermediate nodes* represent unary operators (e.g., the sine function) or binary operators (e.g., arithmetic operators, such as multiplication and division), resulting in one or two arguments (represented as child nodes), respectively. The *root node* represents the final expression element of the tree, which may be either an operator, parameter, or variable, depending on the depth of the tree.

In addition, subtrees within expression trees are called *elementary trees* and are the smallest possible expression for a given node, containing at most one operator. For input variables and constants, the elementary tree is a single node. For unary or binary operators, an elementary tree is the operator node with its children variables and constants.

---

[1]also see Miyazaki et al.[28] for an application of a symbolic regression model, AI-Feynman, to psychology.

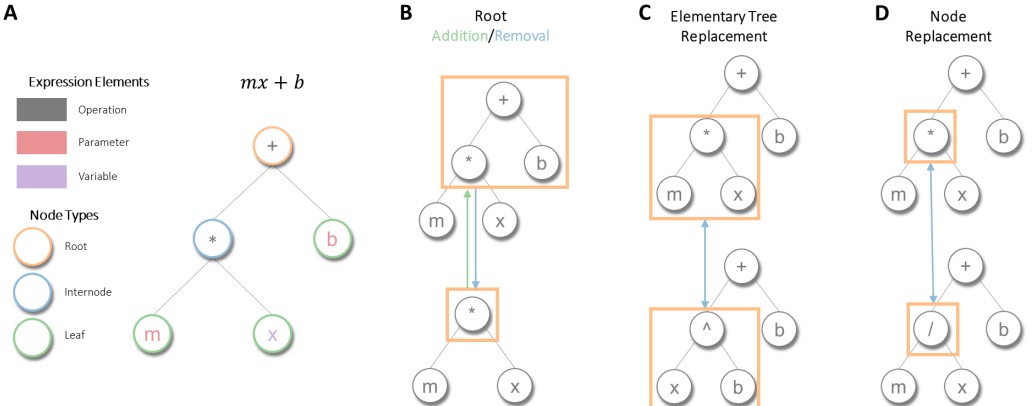

Figure 1: **A**: Mathematical expression trees. Nodes correspond to operations, parameters, or variables. Edges link operations (parent nodes) with their respective arguments (child nodes). Trees can be mutated in three ways. **B**: a *root addition* replaces the current root with an elementary tree, where the root becomes a child node of the new root. A *root removal* removes the root, along with all but one of its children nodes, which becomes the new root node. **C**: An *elementary tree replacement* replaces one of the leaves with an elementary tree. **D**: A *node replacement* replaces any node in the tree with another node with the same number of inputs and outputs.

## 2.2 Expression Tree Search

BMS uses a Markov Chain Monte Carlo (MCMC) approach to search the space of mathematical expressions that best describe the relationship between the input variables and the output variable. There are three sampling actions, each resembling a different tree modification : (1) add or remove the root function from the tree (Figure 1B); (2) replace one of the elementary trees at the base of the tree (Figure 1C); (3) replace any one of the nodes within the tree with another node with the same number of inputs (Figure 1D).

For each tree modification, BMS evaluates the new expression tree on its *description length*, which balances how well the expression fits the data with how well it represents the prior distribution. The description length has no tractable formula, but multiple approximations exist [29, 30]. BMS uses the following approximation [29]:

$$D.L.(f_i) \approx \frac{1}{2T} BIC(f_i) - \log[p(f_i)] \tag{1}$$

where $p(f_i)$ is the prior for the entire expression tree, and BIC is the Bayesian Information Criterion. $T$ corresponds to a temperature value used for parallel tempering. The BIC is calculated as follows:

$$BIC(f_i) = k\log(n) - 2\log[\hat{L}(f_i)] \tag{2}$$

for number of parameters, $k$, number of data samples, $n$, and log-likelihood, $\log[\hat{L}]$. BMS further uses an approximation of the log-likelihood [11]:

$$\log[\hat{L}(f_i)] \approx \frac{n}{2}(1 + \log(2\pi) + \log(MSE^*(f_i))) \tag{3}$$

where $MSE^*$ is the mean squared error of model $f_i(\theta^*)$, with parameters $\theta_i$ fit to data D. Combining equations 1, 2, and 3 gives us the compact loss function used by BMS:

$$D.L.(f_i(\theta^*)) \approx \frac{1}{2}\log\left[\frac{n^k}{2\pi p(f_i)MSE(f_i(\theta^*))}\right] - \frac{n}{2} \tag{4}$$

BMS samples all operations with equal probability, with the set of operations considered limited to those included in the priors. Once evaluated, a new expression is selected if it satisfies Metropolis'

Rule [31]—an algorithm that ensures MCMC samples from the posterior distribution in a statistically unbiased way.

### 2.2.1 Priors

Priors of an equation tree, $p(f_i)$, are calculated as the sum of the priors of its operations, $p(o_i)$:

$$p(f_i) = \Sigma_{o_i \in f_i} [p(o_i)] \tag{5}$$

The priors of individual operations are determined by a back-propagation method that fits the priors to average frequencies of the operations found in the scraped equations, outlined by Guimerà et al., 2020 [11].

### 2.2.2 Parallel Tempering

MCMC is subject to local minima when traversing the space of expressions. BMS addresses this with a parallel tempering search strategy [11] built into its loss function (eq. 1). Parallel tempering involves considering multiple expression trees in parallel, each held at a distinct temperature. At higher temperatures, the BIC is weighted less heavily, discouraging overfitting. This allows for larger changes to the expression at higher temperatures and smaller changes at lower temperatures. Between sampling steps, BMS evaluates pairs of expressions and assigns a lower temperature to ones that fit better. After training is complete, the expression at the lowest temperature, $T = 1.0$, is chosen as the best candidate model. At this temperature, the loss function most accurately approximates the minimum description length of the expression tree.

## 3 Experiment 1

In Experiment 1, we sought to examine the performance of BMS in recovering four psychological models of human information processing. Here, we evaluate the performance of BMS relative to traditional modeling approaches in Psychology, namely a linear/logit regression and a black-box neural network. In doing so, we examine the impact of noise on the recovery.

### 3.1 Methods

#### 3.1.1 Psychological Models of Human Information Processing

We evaluated the performance of BMS in recovering psychological models of human information processing using four ground truth models.

**Steven's Power Law** Steven's power law describes the relationship between a stimulus's intensity $x$ ($\texttt{range} : [0.01, 5.00]$ with 100 equally spaced datapoints) and its perceived magnitude $y$. According to this law, humans are less sensitive to changes in high-intensity stimuli compared to low-intensity ones, leading to a power-law relationship between stimulus intensity and perceived magnitude:

$$y = x^{\alpha}$$

where $\alpha = 0.80$, resulting in diminishing effects of increases in stimulus intensity.

**Weber-Fechner Law** The Weber-Fechner law quantifies the minimum change in a stimulus required to be noticeable. Similar to Steven's power law, the greater the intensity of a stimulus, the larger the change needed to be perceivable. This relationship is hypothesized to be proportional to the logarithm of the ratio between the two stimuli:

$$y = \log\left(\frac{x_1}{x_2}\right)$$

where $x_1$ ($\texttt{range} : [0.01, 5.00]$ with 100 equally spaced datapoints) is the intensity of a physical stimulus (e.g., the luminosity of a lamp), $x_2$ ($\texttt{range} : [0.01, 5.00]$ with 100 equally spaced datapoints) is a reference stimulus (e.g., the luminosity of a background light), and $y$ is the perceived stimulus intensity (e.g. the perception of the lamp's luminosity).

**Shepard-Luce Choice Rule** The Shepard-Luce choice rule, as adapted in [32], posits that the likelihood of an individual assigning a target object, represented as $x$, to a specific response category, represented as $i$, is proportional to their psychological similarity $\eta_i(x)$. Here, we considered a version of the model that computes the probability of assigning the target object $x_1$ to one of two response categories, given a distractor object $x_2$:

$$y = p(\text{``}x_1 \text{ is perceived as category 1''}) = \frac{\eta_1(x_1)\cdot\alpha}{\eta_1(x_1)\cdot\alpha + \eta_2(x_1)\cdot\alpha + \eta_1(x_2)\cdot(1-\alpha) + \eta_2(x_2)\cdot(1-\alpha)}$$

where $\alpha = 0.8$ is an attentional bias toward processing the target object $x_1$, and variables $\eta_i(x_j)$ (`range`: $[0.125, 10.00]$ with 8 equally spaced datapoints) are the psychological similarity between object $x_j$ and category $i$.

**Exponential Learning** The exponential learning equation is one of the standard equations to characterize improvements on a task as a function of task practice [33, 34]. According to this law, the performance on a task $y$ scales as a function of time spent on the task $x$ (`range`: $[1, 100]$ with 100 equally spaced datapoints), as follows:

$$y = \beta - (\beta - x_0) * e^{\gamma * x}$$

where $x_0$ (`range`: $[0, 0.50]$ with 100 equally spaced datapoints) is the initial performance on the task, $\beta = 1.00$ is the maximum (asymptotic) performance on the task, and $\gamma = 0.03$ is the learning rate.

### 3.1.2 Benchmark Models

In most cases, psychologists rely on simple statistical models (such as linear or logistic regression) to examine relationships between experimental variables and observed outcomes.

**Logit Regression**

Our first benchmark model is a logit regression, a linear regression adapted for choice probabilities via a logit (inverse sigmoid) transformation of the data. The logit transformation maps values from 0 to 1 onto $-\infty$ to $\infty$. The data is then fit to a linear regression model, with interaction terms modelled by a second-order polynomial features model. For each ground-truth, all experimental factors and their interactions are included as regressors, while the observations obtained from each ground-truth are considered the regressands. Thus, this model has relatively few coefficients, making it interpretable.

**Multi-layer Neural Network**

Our second benchmark model is a 3-layer neural network. We chose this "black-box" model as a complement to the linear regression as it can capture non-linear relationships. The network model is composed of three hidden layers with 32, 16, and 32 units, respectively, each with a tanh activation function, followed by a linear output layer. A softmax activation function is added to the output layer for data generated by the Shepard-Luce choice rule, to model outcomes as choice probabilities. Training was performed with a learning rate $l_r = 0.0001$, using a cosine-annealing learning rate scheduler, and an ADAM optimizer, for 5000 epochs.

### 3.1.3 Simulation Procedure

We began by generating datasets that corresponded to four ground truth models. Gaussian noise was added to each ground-truth model, spanning seven noise levels measured in units of standard deviations $\sigma$: 0.010, 0.025, 0.050, 0.100, 0.250, 0.500, and 1.000. Higher values within this range were indicative of greater variability found in real-world data. This resulted in 28 datasets, each with the number of datapoints contingent on the range of the corresponding ground-truths independent variables (see 3.1.1). These data were then randomly divided into 80% training and 20% testing subsets. We employed BMS, a logit regression, and a neural network on each dataset, conducting the recovery process 20 times. We here used the priors provided by Guimerà et al., 2020 [11].

### 3.1.4 Model Evaluation

First, as a proof-of-concept, we evaluated the ability for BMS to recover psychological models of information processing in ideal conditions. Specifically, we provided BMS with all datapoints per model with no noise, and let it run for 6000 epochs with a total of 30 parallel-tempered trees.

Second, we evaluated the fit of each model using mean squared error (MSE), log-likelihood, Bayesian information criterion (BIC), and minimum description length (MDL). The mean squared error was computed by contrasting model predictions against test observations, the log-likelihood was derived directly from the mean squared error, the BIC was calculated using log-likelihood along with information about the number of samples and parameters (eq. 2), and MDL was determined by combining the BIC with the priors (eq. 1). Third, we further investigated BMS's performance and ability to derive interpretable models by tallying whether it recovered the correct expression, taking count of all models that were structurally equivalent to the ground truth model. We rounded all parameters to the nearest significant value—2.7182 became e, 0.25001 became 0.25, etc.—if they were within 5% of the absolute value of the significant value. Significant values include integers and the parameter values included in the ground truth equation.

## 3.2 Results

Indeed, BMS was able to recover all models in ideal conditions (Table A1). Additionally, we found that BMS consistently outperformed both the logit regression and the neural network across the majority of noise levels for all evaluation metrics (Figures 2 and A1). Specifically, for both Steven's power law and exponential learning, BMS outperformed both benchmark models for four out of seven noise levels before the models converged in terms of their performance at $sigma = 0.250$. For Weber-Fechner law and Shepard-Luce choice rule, BMS outperformed both benchmark models for six out of seven noise levels before the model performances converged at $sigma = 1.000$. In no instances did either of the benchmark models outperform BMS.

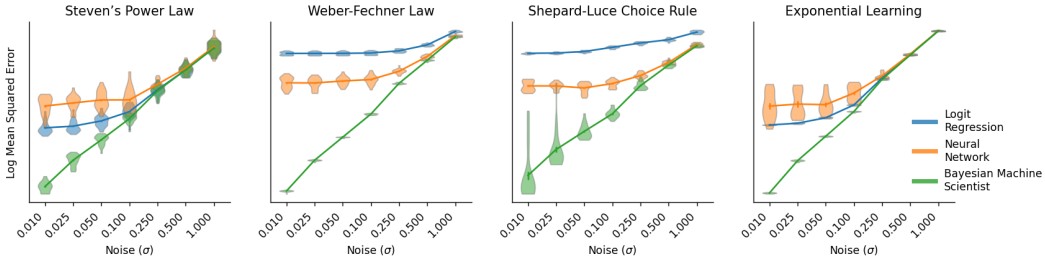

Figure 2: Log-transformed mean squared errors of BMS, logit regression, and neural network for four synthetic models of human information processing across noise levels. Distributions represent the full range of data, lines represent the means, and error bars represent standard error.

To assess the interpretability of the equations produced by BMS, we investigated the degree to which it could recover each psychological model of human information processing. We found that BMS was capable of fully recovering the ground-truth models of interest, albeit with a decline as noise levels and the size of the expression tree increased (Figure 3, also see Table A2 for example expressions produced by BMS). To illustrate, when dealing with our simplest ground truth model, Steven's power law, BMS consistently achieved full recovery under three out of seven noise levels and maintained a 45% recovery rate at the most challenging noise levels. With more complex ground truth models, like exponential learning, BMS demonstrated a 60% recovery rate at the lowest noise levels but only 10% recovery at the highest noise levels.

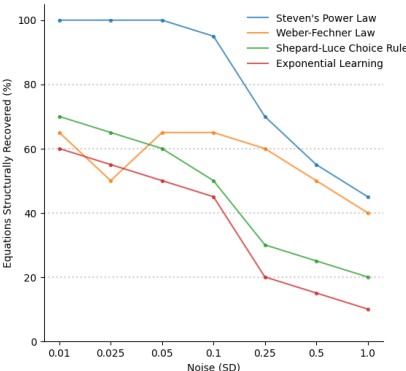

Figure 3: The percentage of times that BMS structurally recovered the models for each ground-truth model of information processing across noise levels.

## 4 Experiment 2

One advantage of BMS is that it allows the incorporation of prior knowledge in the search process as a way of addressing vast search spaces. In Experiment 2, we investigated whether informed priors would aid

BMS in model recovery against a benchmark uniform prior. Specifically, we derived four priors from different research domains ranging in their specificity to psychology.

## 4.1 Methods

### 4.1.1 Scraping Priors

BMS uses priors as a method of estimating the plausibility of a generated expression, and thus its possibility of being selected as the best model. The prior distribution contains the number of times each operator and function appeared *per equation*.

We created four distributions of informed priors by webscraping equations from Wikipedia pages using our open-source package *Equation-Scraper*. *Equation-Scraper* accumulates equations from links to a certain depth and then parses scraped expressions via expression trees. We explored four research domains that ranged in their specificity to psychology—namely, cognitive psychology, cognitive science, neuroscience, and materials science, ordered from most to least related. We used a search depth of two meaning that we investigated all links within the corresponding category page, links within these links, and finally, links within these sublinks. For example, the first path of the cognitive psychology domain was *Cognitive Psychology → Cognitive Psychologists → American Cognitive Psychologists*. We then extracted equations from all levels of these links for parsing. For each scraped expression, we tallied the following operators and functions: $+, -, *, /, x^2, x^3, \sqrt{x}, e^x$, $\sin, \cos, \tan, \text{asin}, \text{acos}, \text{atan}, \sinh, \cosh, \tanh, \log, \text{abs}, \max, \min$.

This process resulted in four informed sets of priors with the count of instances of each operator and function across all expressions (Figure A2). We then divided these counts by the number of equations parsed (Materials Science: 9,581, Neuroscience: 641, Cognitive Science: 609, Cognitive Psychology: 75), which was provided to BMS. Additionally, we created a benchmark uniform prior that contained all operators and functions across all informed priors but with equal occurrences.

### 4.1.2 Simulation Procedure and Model Evaluation

We used the same datasets generated from the ground-truth models in Experiment 1, with the 80%-20% train-test subsets, for the four ground-truth models at the noise level of $\sigma = 0.0250$. We then employed BMS using the four informed sets of priors, and one benchmark uniform prior, as described above. We conducted the recovery process 15 times, and evaluated performance using mean squared error, log-likelihood, BIC, and MDL. BMS's performance and interpretability was also assessed as the percentage of times it recovered structurally equivalent expressions to the ground truth models.

## 4.2 Results

Integrating informed priors into the BMS framework robustly improved performance for only one of the four models of human information processing—namely, exponential learning (Figures 4, A3). The priors for the remaining three models performed equally to the uniform prior. When assessing the interpretability of the equations that BMS produced, similar findings—i.e., a positive impact only with exponential learning—were found when observing the percentage of structurally equivalent models to the ground truth across priors (Figure 5).

## 5 Discussion

Our study aimed to assess the effectiveness of using the Bayesian Machine Scientist (BMS) for the recovery of psychological models of human information processing. We observed that BMS consistently outperformed two benchmark models, namely a logit regression and a neural network, across the four synthetic models. This assessment involved systematically manipulating the noise levels in our simulations, with higher noise levels approximating real-world data. BMS exhibited superior performance compared to the benchmark models for the majority of noise levels tested and demonstrated at least equivalent performance when considering the highest levels of noise. Importantly, there was no instance where BMS was surpassed by the benchmark models, suggesting that BMS may serve as a viable method for discovering psychological models of human information processing and that it may outperform traditional modeling approaches in psychology.

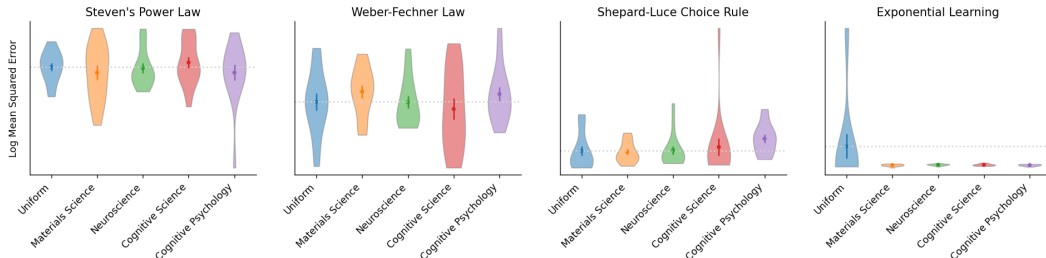

Figure 4: Log-transformed mean squared errors of BMS for four informed priors and one benchmark uniform prior across four models of human information processing. Distributions represent the full range of data, points represent the means, and error bars represent standard error. Dotted horizontal line represents the outcome of the uniform prior for comparison.

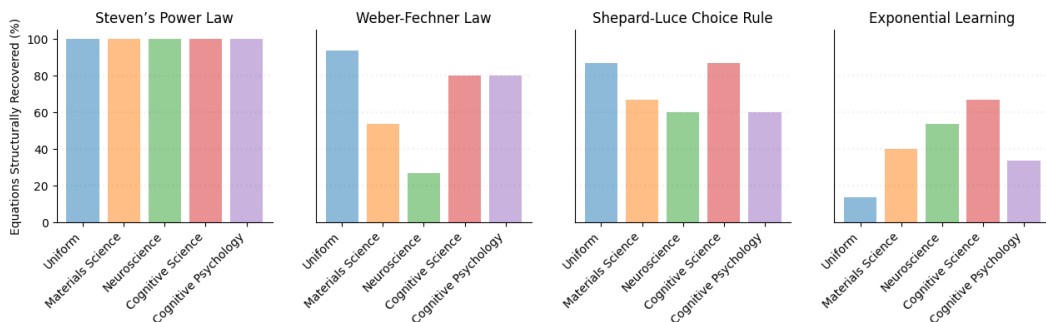

Figure 5: The percentage of times that BMS structurally recovered the models for each prior across the four ground-truth models of information processing.

We proceeded to examine whether the inclusion of priors could enhance BMS's ability to recover these models of human information processing. We explored four priors sourced from research domains, each progressively more specific to psychology, and compared them to a standard, uniform prior. Our investigation revealed that the use of informed priors led to an improvement in BMS performance for one of the four models of human information processing compared to the uniform prior. These findings illustrate that informed priors can be useful in increasing BMS performance in certain contexts, but it is unclear why exponential learning was the only model impacted. Thus, further investigations, perhaps on a wider array of ground-truth models, priors, and noise levels will be necessary to understand the true impact of informed priors on BMS model recovery.

While our findings align with prior research on BMS's effectiveness for model recovery [11] and partly with its enhancement when using informed priors [35–37], several unresolved questions remain. Our study was simulation-based and future work should validate our findings using real-world data to assess BMS's generalizability. Further, our comparisons were limited to a small subset of benchmark models—future research could extend this assessment to a wider array of common psychological modelling techniques and model discovery algorithms (e.g., symbolic regression [7–9], differentiable architecture search [10]). Next, although BMS outperformed benchmark models, it did so with increased computational costs and modelling time. Understanding its computational constraints relative to other models would be critical. In addition, there exists room to optimize BMS, such as parallel processing, to enhance its efficiency. Lastly, the examined priors offered limited information, focusing on operation and function frequency across expressions. Expanding this research with more informative priors, including conditional operations, would be beneficial.

In sum, we demonstrated the effectiveness of BMS in recovering psychological models of human information processing. BMS consistently outperformed benchmark models across various noise levels, affirming its efficacy. Additionally, the inclusion of informed priors improved BMS performance for one of the four models. We thus encourage psychology researchers to consider BMS for automated scientific discovery, yet caution that integrating informed priors may not be robustly impactful.

## Acknowledgments and Disclosure of Funding

S.M. was supported by Schmidt Science Fellows, in partnership with the Rhodes Trust, and all authors were supported by the Carney BRAINSTORM program at Brown University.

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

# A   Appendix

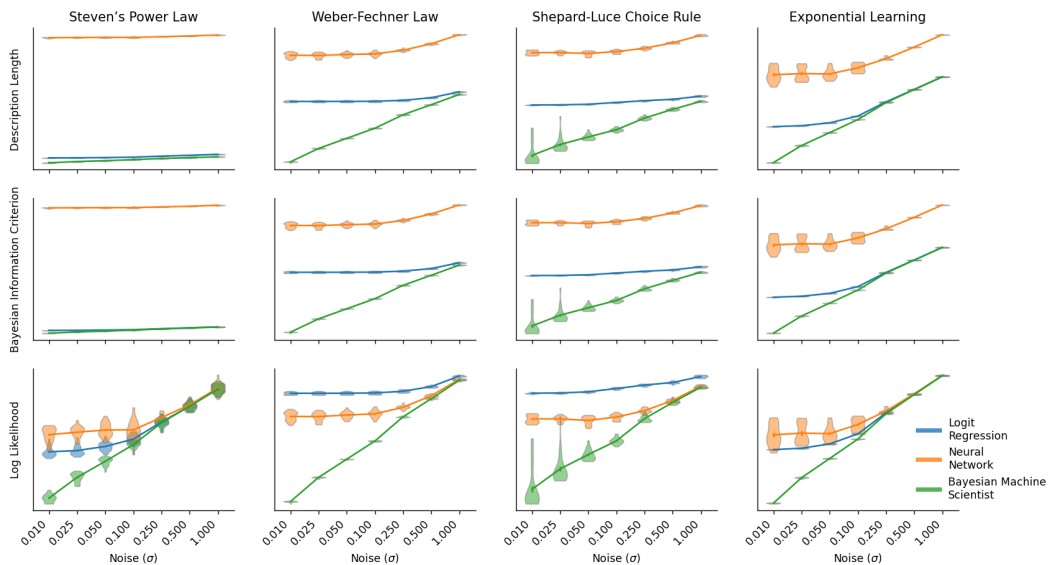

Figure A1: Description length, Bayesian information criterion, and log likelihood of BMS, logit regression, and neural network for four synthetic models of human information processing across noise levels. Distributions represent the full range of data, lines represent the means, and error bars represent standard error.

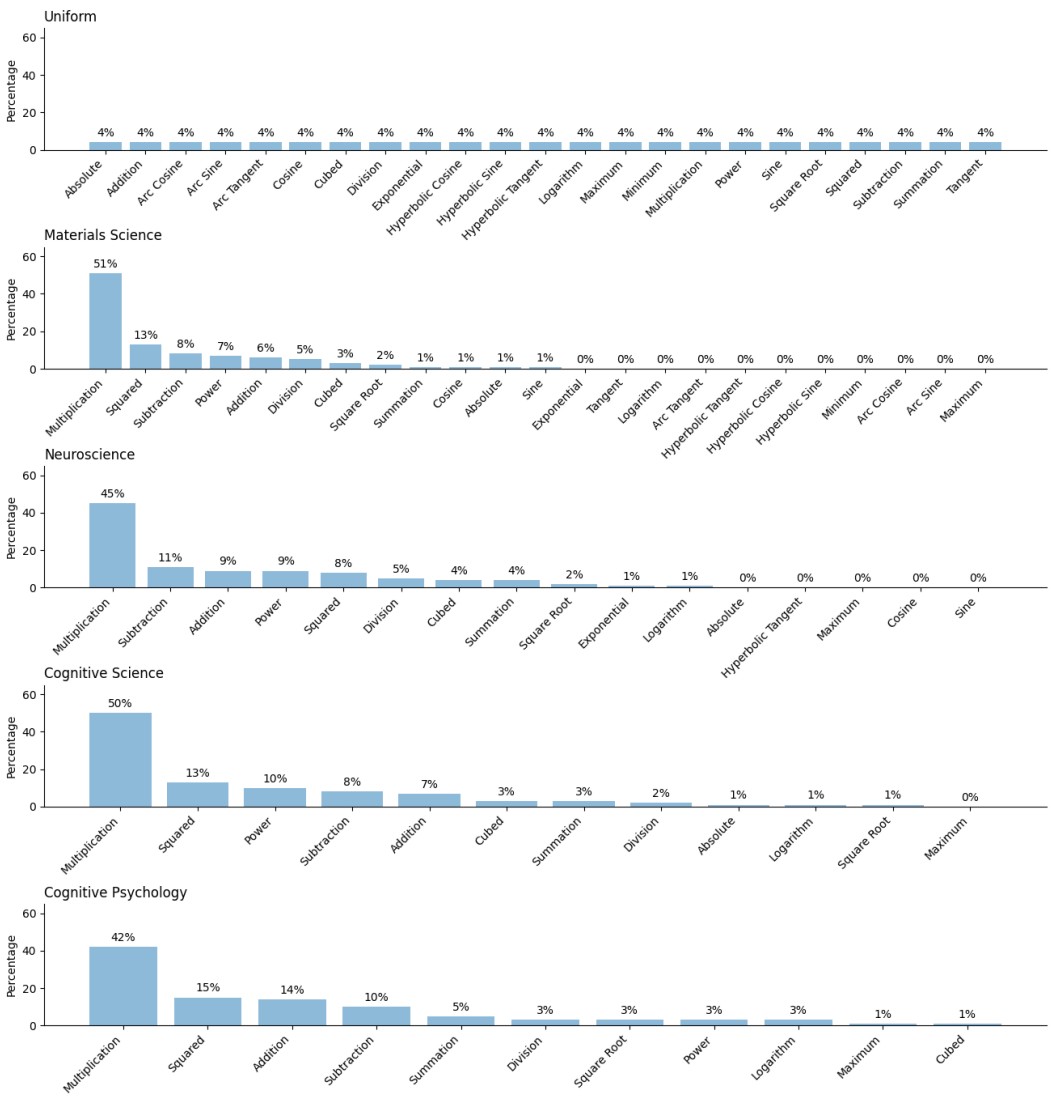

Figure A2: Percentages of the frequency of each operator and function for a benchmark uniform prior and four informed priors of materials science, neuroscience, cognitive science, and cognitive psychology. Operators and functions are included within each prior if they were encountered at least one time. Operators and functions with 0% indicate a value greater than 0% but smaller than 0.5%

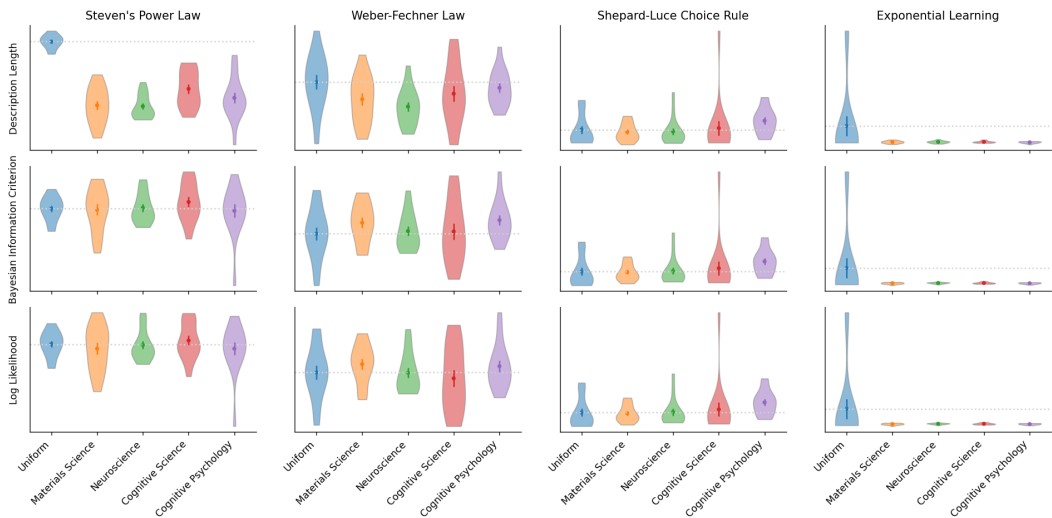

Figure A3: Description length, Bayesian information criterion, and log likelihood of BMS for four informed priors and one benchmark uniform prior across four models of human information processing. Distributions represent the full range of data, points represent the mean, and error bars represent standard error. Dotted horizontal line represents the outcome of the uniform prior for comparison.

Table A1: Proof-of-concept equation recovery using BMS in ideal conditions. Red expressions represent the ground truth; black expressions represent the raw outputs of BMS; blue expressions represent the simplified versions of the BMS expressions. Although the simplified versions of BMS expressions did not always exactly match the ground-truth expressions, they were always structurally equivalent to them. As such, BMS recovered all models given the 5% error margin threshold for parameter values. BMS was prone to over-complicating the equation. Theoretically BMS will simplify the model, given enough training time.

**Steven's Power Law**

$$\textcolor{red}{x^{0.8}}$$

$$x^{0.8}$$

$$\textcolor{blue}{x^{0.8}}$$

**Weber-Fechner Law**

$$\textcolor{red}{\log(\tfrac{x_1}{x_2})}$$

$$\log(\tfrac{x_1}{x_2}) + x_2^{2.19} \cdot 0.0^{22.81}$$

$$\textcolor{blue}{\log(\tfrac{x_1}{x_2})}$$

**Shepard-Luce Choice Rule**

$$\textcolor{red}{\frac{x_1 \cdot 0.8}{x_1 \cdot 0.8 + x_2 \cdot 0.8 + x_3 \cdot (1 - 0.8) + x_4 \cdot (1 - 0.8)}}$$

$$\frac{x_1}{x_1 + (((x_4 + x_3) + (-(1.0) + 1.0)) \cdot 0.25)) + \frac{x_2}{1.0}}$$

$$\textcolor{blue}{\frac{x_1}{x_1 + x_2 + x_3 \cdot 0.25 + x_4 \cdot 0.25}}$$

**Exponential Learning**

$$\textcolor{red}{1.0 - (1.0 - x_0) * e^{0.03 * x}}$$

$$(((((-1.0 \cdot 0.51) \cdot ((0.97^{0.97 + (x^{1.0})}) \cdot (1.0 + (1.0 + ((x_0 + x_0) \cdot -1.0))))) + 1.0) \cdot 1.0)$$

$$\textcolor{blue}{1.0 - 0.51 \cdot (0.97^{0.97 + x} \cdot (2.0 - 2.0 \cdot x_0))}$$

Table A2: Featured equations recovered by BMS for each psychological model at noise level 0.025 from Experiment 1. Red equations represent the ground truth of the corresponding psychological model of information processing.

*Steven's Power Law*

$$x^{0.8}$$

$$x^{-0.2} \cdot x$$

$$(1.23 \cdot (x \cdot 0.81))^{0.81}$$

$$0.02 + (x \cdot 0.98)^{0.81}$$

$$x^{0.8} \cdot (0.8^2 \cdot 1.55)$$

*Weber-Fechner Law*

$$\log\left(\frac{x_1}{x_2}\right)$$

$$\log(x_2 \cdot x_1^{-1.0})$$

$$\log\left(\frac{x_2}{x_1 \cdot 1.0}\right)$$

$$\left(\left(\frac{x_2}{x_1}\right)^{0.0} - 1.0\right) \cdot 7177.24$$

$$\left(\left(\left(\frac{x_2}{x_1}\right)/4261.66\right)^{-0.0} \cdot -4253.3\right) + 4261.66$$

*Shepard-Luce Choice Rule*

$$\frac{x_1 \cdot 0.8}{x_1 \cdot 0.8 + x_2 \cdot 0.8 + x_3 \cdot (1 - 0.8) + x_4 \cdot (1 - 0.8)}$$

$$\left(\frac{x_1}{-1.03 + 0.25 \cdot x_3 + (x_2 + 0.25 \cdot x_4 + x_1 + x_1^{0.04})}\right) \cdot 1.01$$

$$\left(\sqrt{0.25 \cdot \frac{x_1 \cdot 3.99}{(3.99 \cdot x_3 \cdot 0.06) + (x_1 - 0.01) + \left(x_2 + \frac{x_4}{3.99}\right)}}\right)^{2.0}$$

$$\left(\left(\tanh\left(\exp\left(\frac{x_1}{0.33 \cdot (x_3 + (x_1 \cdot 0.33) + (x_4 + (0.77 + -0.67 + x_2)) + x_2)^{0.33} \cdot -2.27}\right)\right)\right) + -1.01\right)^3$$

$$\left(\left(\left(\left((x_3 + ((x_4 + x_2) \cdot 0.25)) + (0.02 + x_1)\right) \cdot \log\left(0.61^2\right)\right) \cdot 1.01\right) + 0.0\right) \cdot x_1$$

*Exponential Learning*

$$1.0 - (1.0 - x_0) * e^{0.03 * x}$$

$$1.0 + (0.97^x \cdot (-(-x_0) + -1.0))$$

$$\frac{(0.81 \cdot -1.23 + x_0)}{(e^{0.81} \cdot 1.23)^{x+x} + 1.0}$$

$$\tanh\left(\left((0.29 + x_0)^{1.69} + x \cdot 0.02\right)^{0.75} - 0.21\right)$$

$$\cos\left(\cosh\left(\frac{8.63}{\log\left(1.23^{\left(-2.25 \cdot 8.63 + x_0 + \left(e^{1.23}\right)^{x_0}\right)} + x\right)}\right) \cdot 1.22\right)/1.02$$

