# OpenReview forum: "Bayesian Machine Scientist for Model Discovery in Psychology"
_NeurIPS.cc/2023/Workshop/AI4Science — NeurIPS2023-AI4Science Poster_

### Official Review · Reviewer_srjk · 2023-10-21
**Valuable study with many questions left unanswered**

**Rating:** 5
**Confidence:** 3

**Review:**

This study focused on using a Bayesian machine scientist algorithm to facilitate interpretable equation discovery in psychology. The authors presented results on simulations of four equations from psychology, and they showed that their model was able to outperform two baselines in fitting the data.

I found this work interesting, particularly the focus on psychology applications as this seems under-explored in the AI for Science community. The authors nicely presented the method and explained the equations on which they were demonstrating their method. However, I have questions still remaining. First, the authors discuss at length the applicability of interpretable elements of the BMS model, but no analysis of this interpretable element is presented in the results. The results seem to center around goodness-of-fit to the simulated data, but are the equations discovered by the BMS algorithm actually matching the underlying scientific model that generated the simulations? It would be useful to show the equations discovered by BMS to validate their similarity to ground-truth equations. Second, the authors only compared to a logistic regression model and a very small MLP. While these models offer good comparisons in terms of diversity of ML algorithms, they do not focus on the interpretable elements of the motivation of this work. Since the study is focused on simulated data, I would expect their model to exactly recover the final equation if given enough data, so the authors should try to show this. Finally, as the authors mentioned in the Discussion, there is extensive work in symbolic regression on fitting models that are directly interpretable as mathematical equations to be used in the sciences. It would be helpful to benchmark against these methods, or at least qualitatively compare their method to existing work in symbolic regression. Finally, I agree that psychology is understudied in AI for science, but it seems as if symbolic regression has already been applied to psychology data before [1], thereby weakening the novelty of this study.

Overall, I believe this work is slightly below the acceptance threshold. I understand the value of highlighting equation discovery in psychology and the need for more development in the AI community, but the authors have much to do to contextualize their work to previous work as well as analyzing the core interpretability of the method. Since the focus of the study is on interpretability of these methods, I think the work would greatly benefit from an analysis in this regard.

[1] Miyazaki et al., Frontiers of Artificial Intelligence, 2023 (https://www.ncbi.nlm.nih.gov/pmc/articles/PMC9911656/)

---

### Official Review · Reviewer_sDZw · 2023-10-25
**Applies Bayesian inference-based method to psychology model discovery**

**Rating:** 7
**Confidence:** 1

**Review:**

This paper explores the idea of applying a Bayesian inference-based method named BMS to derive psychological models. The experiments seem to be solid: It compares BMS with several baselines and demonstrated better performances. Moreover, it compares different types of priors effect on BMS since it allows the incorporation of prior knowledge. The paper is generally well-written, and should bring value to the field.

---

### Meta-Review · Area_Chair_EZfs · 2023-10-27

**Recommendation:** Accept (Poster)
**Confidence:** 5

**Metareview:**

**Overview:**
The paper introduces and examines the Bayesian Machine Scientist (BMS) method, which leverages Bayesian inference to derive mathematical models in the context of psychology, particularly for human information processing. Through two experiments, the paper evaluates the utility and robustness of BMS against conventional benchmarks, and also explores the potential enhancement BMS may exhibit with informed priors.

**Strengths:**

1. **Novel Application:** Applying Bayesian inference-based methodologies, especially BMS, for psychological model discovery is relatively novel and has the potential to offer new insights and methodologies to the field.

2. **Solid Experimentation:** The work provides a thorough experimental setup, comparing the BMS approach against standard benchmarks, which is commendable. The comparison across various noise levels also adds depth to the evaluation.

3. **Incorporation of Priors:** The exploration of how informed priors might influence the BMS model is an insightful addition, reflecting the nuances of domain-specific knowledge in model performance.

4. **Well-written Manuscript:** The general quality of the writing and presentation seems to be solid, making the paper accessible and coherent.

**Areas for Enhancement Based on Reviewers’ Feedback:**

1. **Interpretability Analysis:** Reviewer srjk notes the absence of an in-depth analysis on the interpretable aspects of the BMS model. For a work that highlights interpretability as a motivation, this omission might reduce the overall impact of the study.

2. **Model Validation:** Reviewer srjk also raises concerns about the model's capacity to recover the ground-truth equations from simulations. Demonstrating such recovery, especially when using simulated data, would strengthen the paper's claims.

3. **Benchmarking Concerns:** The choice of benchmarks, particularly the omission of symbolic regression approaches, is highlighted as a concern. Adding comparisons against these methods or at least a qualitative discussion regarding them could enrich the paper's discussions.

4. **Novelty Issues:** The potential lack of novelty, considering similar approaches have been previously applied in psychology, needs to be addressed.

**Recommendation:**
Given the innovative nature of the study and its potential implications for the field of psychology, it is recommended that the paper be accepted for a poster presentation. This format will allow for the work to be shared and discussed, while also providing the authors with a platform to address the concerns raised by reviewers, perhaps during the poster Q&A or in subsequent iterations of the work. It is crucial, however, that the authors consider integrating the feedback from the reviewers to further enhance the quality and impact of the paper.